# Mouse-Derived Isograft (MDI) In Vivo Tumor Models II. Carcinogen-Induced cMDI Models: Characterization and Cancer Therapeutic Approaches

**DOI:** 10.3390/cancers11020242

**Published:** 2019-02-19

**Authors:** Janette Beshay, Peter Jantscheff, Thomas Lemarchand, Cynthia Obodozie, Christoph Schächtele, Holger Weber

**Affiliations:** 1In Vivo Pharmacology, ProQinase GmbH, Breisacher Str. 117, 79106 Freiburg, Germany; Janette_Beshay@web.de (J.B.); c.obodozie@proqinase.com (C.O.); c.schaechtele@proqinase.com (C.S.); 2TPL Pathology Labs, Sasbacher Str. 10, 79111 Freiburg, Germany; lemarchand@tpl-path-labs.com

**Keywords:** mouse tumor models, experimental cancer, mouse-derived isografts (MDIs), carcinogen-induced tumors, therapy, immune checkpoint inhibitors, immunocompetent animals, syngeneic

## Abstract

In this second study, we established syngeneic in vivo models named carcinogen-induced mouse-derived isografts (cMDIs). Carcinogen-induced tumors were obtained during short-term observation (3–9 months) of CBA/J mice treated with various administration routes with 3-methylcholanthrene (MCA) or *N*-methyl-*N*-nitrosourea (MNU) as carcinogens. During necropsy, primary tumors and suspicious tissues were assessed macroscopically and re-transplanted (in PDX-like manner) into sex-matched syngeneic animals. Outgrowing tumors were histologically characterized as either spinocellular carcinoma (1/8) or various differentiated sarcomas (7/8). Growth curves of four sarcomas showed striking heterogeneity. These cMDIs were further characterized by flow cytometry, RNA sequencing, or efficacy studies. A variable invasion of immune cells into the tumors, as well as varying expression of tyrosine kinase receptor, IFN-γ signature, or immune cell population marker genes could be observed. Immune checkpoint inhibitor treatment (anti-mPD-1, anti-mCTLA-4, or a combination thereof) showed different responses in the various cMDI models. In general, cMDI models are carcinogen-induced tumors of low passage number that were propagated as tissue pieces in mice without any tissue culturing. Therefore, the tumors contained conserved tumor characteristics and intratumoral immune cell populations. In contrast to the previously described spontaneous MDI, carcinogen induction resulted in a greater number of individual but histologically related tumors, which were preferentially sarcomas.

## 1. Introduction

In the accompanying paper, the establishment and characterization of a new type of experimental in vivo cancer model, so-called mouse-derived isografts (MDIs) from spontaneous tumors (sMDIs), have been reported for the first time [1]. In the first part of this study, we introduced the establishment of nine histopathologically different sMDI tumor models in various mouse strains of both sexes, i.e., on different H-2 MHC class I haplotype backgrounds. The various sMDIs displayed heterogeneous take rates and growth curves. Preliminary efficacy data of immune checkpoint inhibitor (ICPI) treatment with anti-mPD-1, anti-mCTLA-4 antibodies, or a combination thereof, as well as treatment with the chemotherapeutic gemcitabine, indicated different sensitivities to the different types of treatment. Additionally, flow cytometric analysis revealed variable tumor infiltration by immune cells compared to commonly used murine, cell line-based syngeneic standard tumor models [2]. RNA sequencing analysis showed different genetic modifications in the expression of tyrosine kinase receptors, IFN-γ-signature, or immune cell population markers comparing various MDI models [1]. The new models increased the quantity and quality of available syngeneic in vivo tumor models, since we generated novel primary adenocarcinomas, lymphomas, and histiocytic sarcoma/histiocyte-associated lymphomas. These new spontaneous tumor models enable studying the causes and molecular mechanisms of tumor development, as well as new therapeutic approaches, especially regarding the interaction between the immune system and the tumors.

In this second study, we established and characterized MDI cancer models by the induction of tumor formation by treatment with carcinogens 3-methylcholanthrene (MCA) or *N*-methyl-*N*-nitrosourea (MNU). The generated models were named carcinogen-induced MDI (cMDI) tumors. The goal of this study was to create more histopathologically closely related tumor entities derived from one mouse strain. Similar to sMDI tumors, cMDI tumors display low passage numbers and were propagated only in vivo as tissue pieces in syngeneic mice without any tissue culturing (in a patient-derived-xenograft (PDX)-like manner) resulting in rather conserved original tumor characteristics and immune cell populations [1]. Carcinogen induction resulted in sarcomas, in most cases, in H-2^k^ CBA/J mice of both sexes.

The approach of chemical induction of syngeneic mouse tumors by carcinogens has the advantage of inducing tumors faster and in larger numbers than it would take to wait for spontaneous tumors. However, some disadvantages are strain-specific differences in carcinogen sensitivity [3] and that the tumors are mostly sarcomas, which is a rarer tumor entity than adenocarcinomas, carcinomas, or lymphomas in patients and experimental animals [4]. Another difference to spontaneous tumors is a higher immunogenicity of carcinogen-induced tumor cells often caused by the expression of tumor-associated transplantation antigens (TATA) [5,6,7]. However, as demonstrated in rat MCA-induced fibrosarcomas, not all tumors (but about 75%) actually display such antigens [8].

In this study, we outline the establishment and characterization of tumors that are the result of carcinogen administration in H-2^k^ CBA/J mice of both sexes, and assess their use for future research.

## 2. Results

### 2.1. cMDI—Establishment History

For the establishment of cMDI, 12 female and 16 (originally, but eventually 13, since 3 animals died within 1–2 days after i.p. injection) male CBA/J mice were treated with MCA or MNU (Table 1).

Due to ethical reasons, i.e., critical weight loss, bad general condition, or externally detectable tumor growth, a necropsy of the animal was performed (Figure 1). The first necropsy was about two months after subcutaneous injection of MCA in a male CBA/J mouse (JA-2044) after developing a palpable subcutaneous tumor, and outgrowth of directly re-transplanted or frozen tumor pieces varied between 39 and 113 days after re-transplantation (Figure 2, Table 2). After about three months, a subcutaneous tumor was detected in another male mouse (JA-2041) treated subcutaneously with MCA. Also, during necropsy, suspicious prostate tissue was found and re-transplanted into syngeneic mice. Since only re-transplanted fresh or frozen subcutaneous tumor tissue led to stable outgrowth between 27 and 52 days (Figure 2, Table 2), collected frozen prostate tissue was not further pursued.

Approximately one month later, a growing tumor was detected in the right hind leg of a female CBA/J mouse (JA-2019) which received subcutaneous administration of MCA. During necropsy, an enlarged and strongly vascularized axillary lymph node was also found but not further processed. Re-transplanted frozen tumor tissue grew well between 24 and 35 days in syngeneic mice (Figure 2, Table 2). Another month later, another animal had to be sacrificed due to ethical reasons after a small tumor on the right flank of a female CBA/J mouse (JA-2011) was ulcerated. This mouse received subcutaneous administration of MNU, and the re-transplanted frozen tumor grew between 21–30 days (Figure 2, Table 2). About two weeks later, a fourth tumor was removed from the right hind leg of a male after intramuscular injection of MCA (JA-2042), which grew between 28–35 days (Figure 2, Table 2).

In the following months, three other cMDI tumors were detected (Figure 2, Table 2). They were established from a male animal with subcutaneous MNU treatment (JA-2034), a female mouse with subcutaneous MCA treatment (JA-2017), and a male CBA/J mouse with subcutaneous MCA treatment (JA-2043) with growth between 29 and 55; 14 and 28; and 17 and 36 days, respectively (Figure 2, Table 2). The latter showed invasive growth into the peritoneal wall with massive vascularization (Appendix A). In contrast to other cMDIs, the primary JA-2017 tumor did grow after re-transplantation but only in immunodeficient SCID/bg and not in syngeneic mice (similar to JA-0021 sMDI [1]). This was also the case for SCID/bg-derived, re-transplanted daughter tumors, e.g., 1573-16 (Figure 2, Table 2).

During a total observation time of about 10 months, eight carcinogen-induced (2 × MNU, 6 × MCA) cMDI tumors were finally established, i.e., a stable re-transplantation into syngeneic or immunodeficient (SCID/bg) recipients could be performed using frozen tumor pieces. In most cases, localized cMDI tumors were induced, and only in some cases were suspicious tissues detected. Carcinogen induction (carcinogen and route), animal number, sex, tumor types, and general growth characteristics of all established cMDI models, as well as general histopathological diagnoses (see below), are summarized in Table 2 and Figure 2, Appendix A.

Additionally, tumors or suspicious tissues were detected, and the primary isolates were re-transplanted in nine other mice (s.c. MNU **♀**, s.c. MCA **♀**, s.c. MCA **♀**, p.o. MCA **♀**, p.o. MCA **♀**, i.p. MNU **♂**, p.o. MNU **♂**, p.o. MNU **♂**, and p.o. MNU **♂**). The tumor and tissue pieces were stored frozen until further use. Thus, carcinogen treatment (5 × MNU, 4 × MCA) by various routes (3 × s.c., 5 × p.o., 1 × i.p.) induced malignant growth in a total of 17/28 (i.e., 25 alive) mice. However, in the latter nine cases a stable outgrowth of potential frozen cMDI samples must be verified as the basis for further development.

### 2.2. cMDI—Histological and Pathological Analysis

To characterize the pathological phenotypes of cMDI and to verify the phenotypic stability of tumor tissue after several rounds of re-transplantation, hematoxylin and eosin (HE) staining of the tissue was performed. For this purpose, primary isolates (tumor or suspicious tissue/s) and derived follow-up subcutaneous re-transplants were compared. Photographs of the primary in situ location, typical HE stains of tumor tissues, and primary diagnoses are shown in Figure 2. In contrast to heterogeneous tumorigenicity in spontaneous sMDI models (ADC, HAL/HS, and lymphomas), only one cMDI tumor was characterized as a spinocellular carcinoma, whereas all other established tumors were identified as sarcomas. Single models are described below. More detailed histopathological characterization, as well as larger microphotographs of various cMDI models, are shown in the Appendix A. In addition, a preliminary assessment of the histological analysis of the host stroma reaction and inflammatory response, including the tumor-infiltrating leukocyte index, is shown for some of the models (Appendix A). These findings were confirmed and refined by flow cytometry and RNA-Seq analysis of cMDI JA-2011, JA-2019, JA-2041, or JA-2042 (see below). These first four cMDI models were also used to determine take rates, growth curves, and to perform the first efficacy studies (see below).

JA-2011/0242-17: NOS (not otherwise specified) sarcoma composed of spindle-shaped elongated cells (neoplastic mesenchymal cells) with numerous giant mononuclear or multinucleate giant cells. The tumor is deeply invasive of adjacent tissues. The re-transplanted tumor 0242-17 contains more “anaplastic cells”, hallmark of a more anaplastic “aggressive” sarcoma than the index case (Figure 2, Appendix A).

JA-2019/0174-17: NOS storiform sarcoma composed of interwoven bundles of spindle-shaped elongated cells (neoplastic mesenchymal cells) with minimum anaplasia and clearer and denser areas. They are deeply invasive in both skeletal muscle and adipose tissue (Figure 2, Appendix A).

JA-2041/1607-16: NOS sarcoma composed of interlacing bundles of alternating longitudinal and transverse bundles or long well-differentiated spindle-shaped cells. There is no ischemic necrosis in the tumor of the index case. The homogenous sarcomatous tissue is infiltrating the skeletal muscles. Neoplastic cells exhibit low pleomorphism but have relatively numerous infiltrating lymphocytes and deeply invade into atrophic muscle fibers (Figure 2, Appendix A). The derived established case 1607-16 depicted shows ischemic necrosis indicative of inadequate vascularization, indicative of cancer progression, and with numerous infiltrating lymphocytes.

JA-2042/0124-17: NOS sarcoma with hemangiosarcoma-like differentiation composed of interwoven bundles of spindle-shaped elongated cells with plump nuclei, minimum anaplasia, and clearer and denser areas. Both the source and established tumor either invades adjacent to blood vessels eliciting vascular ectasia or exhibits vascular differentiation, mimicking a hemangiosarcoma. The index tumor also has rare spaces related to single cell degeneration (Figure 2, Appendix A). Notice the excellent phenotypic stability in most of these four tumor samples. The tumor invades the skeletal muscle, and its likely origin is the integument of the leg.

JA-2034/0125-17: NOS sarcoma index case JA-2034 developed within the adipose tissue of the panniculus, which is diffusely invaded by bundles of spindle-shaped cells. The tissue is well-differentiated and of moderate anaplasia. The tissue site of invasion is hypodermis (panniculus). Ischemic necrosis is absent. Neoplastic cells exhibit moderate pleomorphism (no bizarre anaplastic cells) and numerous small infiltrating lymphocytes. The established derived tumor features an increased degree of anaplasia, more mitoses, with a few bizarre giant cells (green arrows) and evidence of limited early ischemic necrosis, indicative of inadequate vascularization and thus selection of more aggressive cells (Figure 2, Appendix A).

JA-2043/0074-17: NOS sarcoma with varying aspects from well-differentiated to moderately anaplastic; no giant cells and weak to moderate leukocyte infiltration. It is deeply invading the skin up to the upper dermis. The tissue shows many anaplastic features (abnormal mitoses, large vesicular nuclei) but no giant multinucleate cells and good phenotypic stability (Figure 2, Appendix A).

JA-2044/1418-16: Interestingly, all other established cMDI tumors were identified as sarcomas. Only JA-2044 and follow-up tumors were characterized as spinocellular (epidermoid) carcinoma, which is an epithelial neoplastic proliferation from the skin characterized by epidermal differentiation and keratin production. It induces intense inflammation with a pronounced lymphoid component as well as fibrosis. It is characterized by cords and nodular elements of keratinocytes with central laminar keratinization, often forming central cysts with occasional horn pearls. The neoplasm entraps a nerve. The majority of the tumor sections examined is composed of desmoplastic and highly inflamed stroma (Figure 2, Appendix A). It exhibits good phenotypic stability.

JA-2017/1573-16: NOS sarcoma varying from well-differentiated to moderately anaplastic; no giant cells. In contrast to the other cMDIs, re-transplanted JA-2017 tumors grow only in SCID/bg, but not in the syngeneic CBA/J strain. However, the diagnosis is identical for both the primary CBA/J and the secondary SCID/bg tumor (Figure 2, Appendix A). Both invaded intracutaneously and peritoneally, with weak to moderate leukocyte infiltration. The established derived tumor exhibits multiple small foci of necrosis, whereas the index tumor does not, indicative of a more aggressive neoplasm (tumor progression).

### 2.3. cMDI—Take Rates and Growth Curves

After model establishment, i.e., a stable outgrowth of frozen tumor pieces, growth curves, and take rates of such transplanted tumor pieces were determined in four cMDI models, JA-2011, JA-2019, JA-2041, and JA-2042 (Table 2, Figure 3). Mean growth times and take rates of the four other models were calculated from the outgrowth of frozen or directly re-transplanted tumor pieces (Table 2).

Comparing growth curves of the four models, JA-2011, JA-2019, JA-2041, and JA-2042, showed variable tumor growth periods of 14–35, 16–63, 25–105, and 25–46 days, and take rates varying from 83%–100% (Table 2, Figure 3) *appearance*, i.e., the earliest time point(s) allowing robust randomization at mean tumor volumes between 40 and 150 mm^3^ in the 12 (or less, depending on the take rate) animals are varying, from 10 to 12; 4 to 7; 7 to 11; and 9 to 13 days, respectively (Table 1, Figure 3). However, the resulting observation period and *treatment time window* (*TTW*) were not simply the difference between *appearance* and the determined tumor growth period. Since a varying quantity of animals in single tumor models had to be sacrificed due to fast tumor growth or ulcerations (ethics), or were found dead for unknown reasons, the number of animals still alive had already critically decreased before the end of the growth periods. Therefore, we defined the so-called *real running time* of the models. *RRT* determines the time difference from implantation to the time point when the remaining animal number reaches ~60% of the starting animal group size.

Thus, it also defines *TTW*, i.e., the maximal time range to treat animals after randomization (*AP*) to the potential study end. This allows for calculation of a realistic study length as well as necessary group size for statistically reliable analysis for testing, e.g., immune checkpoint inhibitors. For the four cMDI sarcoma models, we outlined potential study endpoints, with *RRT*s (and actual *TTW*s) of about 22–30 (12–18), 21–25 (17–18), 25–30 (18–19), or 28–39 (19–26) days (Table 2).

The models showed not only variable growth periods but also varying growth properties. A heterogeneous growth could be observed in the case of three of the four characterized models, JA-2019, JA-2041, and JA-2042 (Figure 3). Only tumor JA-2011 showed more homogeneous growth, but in this model, individual tumor-bearing animals were also lost by ulceration and, later, by fast tumor growth (ethics). In the three other models, a few tumors were slow-growing, while other tumors grew fast or rather moderately. It remains unclear whether these variations in tumor growth reflected differences between re-transplanted tumor pieces or between individual mice, or were caused by chance.

Individual mice of selected models were chosen for further analysis by flow cytometry (JA-2011, JA-2019, JA-2041, and JA-2042), as well as RNA-Seq analysis (JA-2011 and JA-2042). Additionally, enough animals could be included for efficacy studies to investigate antitumor treatment effects in the four cMDI models (see below).

Hence, to also apply the other established cMDI models, e.g., to perform efficacy studies, the exact growth curves and take rates still need to be determined.

### 2.4. cMDI—Flow Cytometric Analysis of Tumor-Infiltrating Leukocytes

Material of re-transplanted and outgrown cMDI was isolated and processed for flow cytometry to analyze various populations of tumor-infiltrating leukocytes. Single cell suspensions were stained with master antibody mixes (T cell panel, myeloid cell panel, macrophage cell panel). The results of flow cytometric analysis for cMDI JA-2011, JA-2019, JA-2041, and JA-2042 are shown in Figure 4A. A quantitative comparison of intratumoral immune cell populations of various cMDI is shown in Figure 4B. Comparing these results with former flow cytometric analyses of seven syngeneic standard in vivo mouse models, MC38-CEA, CT26.WT, LL/2, Clone M3, 4T1, RENCA, or B16.F10 (for details of the models-Appendix A), most striking differences were seen regarding a massive infiltration of CD4^+^ lymphocytes (~3-fold higher) in cMDI JA-2019, JA-2041, and JA-2042, but not in cMDI JA-2011 (Figure 4B). Regarding other immune cell populations (CD8^+^, M1 and M2 macrophages, neutrophils, and M-MDSC), the four models were in the same range as standard syngeneic mouse models but with striking individual variations. However, in Treg cells, an enhanced (2–3-fold) invasion was seen in JA-2019 and JA-2042. Another exception is the 3-fold enhanced invasion of M1 macrophages into JA-2041 and a 1.5-fold higher number of neutrophils found in JA-2011 tumors (Figure 4B).

### 2.5. cMDI—Preliminary Results of RNA-Seq

A comparison of RNA-Seq analysis of sMDI JA-0009, as well as cMDI JA-2011 and JA-2042, was performed, and details are shown in the Appendix A [1]. RNA-Seq analysis was mainly conducted to obtain an overview on target expression for potential new drug development and to identify potential mutations. Additionally, complete RNA-Seq whole transcriptome shotgun sequencing analysis will provide a further tool to localize tumor tissue origin.

In the first step, expression profiles of various genes of three gene families related to tumor malignancy or antitumor immune response were created: tyrosine kinase receptors, immune population markers, and IFN-γ signature, which comprises multiple interferon-responsive genes involved in innate and adaptive immune activities (see accompanying paper [1]).

A direct comparison between individual MDI models was not possible, since FPKM (fragment per kilobase million) values were not determined in simultaneous experiments. However, indirect comparison of individual gene expression based on an internal, low expression reference gene (Appendix A) [1] shows very different individual gene expression patterns of the two histologically related cMDI sarcomas (JA-2011 and JA-2042). Whether this reflects tumor induction by different routes (s.c. versus i.m.) and carcinogens (MNU versus MCA) remains unclear. The RNA-Seq data further support and confirm findings already obtained by other methods. For example, RNA-Seq displayed enhanced Cd4 gene expression in JA-2041 (Appendix A), which confirmed flow cytometry data regarding actual enhanced CD4^+^ T cell tumor infiltration in this cMDI. The details are shown in the Appendix A [1]. A more complete characterization of gene expression of the whole transcriptome shotgun sequencing analysis of further gene families, as well as in further sMDI and cMDI tumor models, will be the matter of subsequent investigations.

### 2.6. cMDI—Efficacy Studies

For further characterization of cMDI models, four efficacy studies were performed using various anti-ICPI antibodies. In a first study, animals implanted with JA-2011 tumor pieces were randomized on day 13 into groups of 12 animals each.

Starting on the day of randomization, animals were treated three times (dotted lines) with anti-mPD-1 antibody or anti-mCTLA-4, as shown in Figure 5. Neither anti-mPD-1 nor anti-mCTLA-4 showed significant effects on tumor growth.

In a second study, JA-2019 tumor pieces were implanted, and mice were randomized on day 7 into groups of 12 mice each. Starting on the day of randomization, mice were treated in a similar schedule with anti-ICPI antibodies. In contrast to JA-2011, a tumor inhibitory effect was induced, which was significant for anti-mPD-1 (** *p* = 0.0064) as well as anti-mCTLA-4 (* *p* = 0.0112) on day 21.

In a third study, JA-2041 tumor pieces were implanted, and randomization took place on day 11. At randomization day, the first treatment with anti-mPD-1 and anti-mCTLA-4 was performed. Although both antibodies reduced tumor growth by about 50% after three treatments, none of the effects were significant when compared on day 30. Interestingly, anti-mCTLA-4-treated tumors showed a very diverse responses: 9 out of 12 tumors displayed reduced tumor volume by about 80–90%, whereas the other tumors did not show any response. It is unclear whether this reflects actual differences between re-transplanted tumor pieces or between individual mice, or is caused by other reasons.

In a fourth efficacy study, using JA-2042 tumor pieces, we were interested in the possible effects of anti-mPD-1, anti-mCTLA-4, or a combination of both antibodies, which may amplify or induce tumor inhibitory effects [9]. After randomization on day 13, groups of 10 mice were treated as described above. The antibody combination led to a significant reduction (** *p* = 0.0025) of tumor volume on day 26. Both single treatments with anti-mPD-1 and anti-mCTLA-4 moderately reduced tumor volume by about 25%, which was not significant. The anti-mCTLA-4 monotherapy seems to result in two effects: strong tumor growth inhibition in a part (5/10) of the tumors, and no effect in others (3/10), similar to the observations within the JA-2041 and JA-2019 tumor models.

## 3. Discussion

The new MDI in vivo cancer models display a model quality and properties unavailable with standard syngeneic tumor models and are therefore closer to actual clinical situation in patients [1]. sMDIs represent outgrowing spontaneous tumors (or metastases) which have overcome the body’s own regulatory mechanisms, as already introduced in the accompanying paper. They are transplantable, i.e., “single step” tumorigenic not only in the primary tumor-bearing animal, but also in other syngeneic, fully immunocompetent hosts without any prior or subsequent additional in vivo or in vitro manipulation [1].

Here, we established syngeneic carcinogen-induced mouse-derived isograft (cMDI) models from once subcutaneously, intravenously, intramuscularly, intraperitoneally, or three times orally with MCA or MNU injected, otherwise untreated, CBA/J mice of both sexes (Table 1). The general characteristics of cMDI are similar to those of sMDI tumors, i.e., primary tumors of low passage number were propagated only in vivo as tissue pieces in syngeneic mice (in a PDX-like manner), resulting in rather conserved tumor characteristics and tumor-infiltrating immune cell populations [1]. However, in contrast to sMDI, the animals for cMDI development have been manipulated by carcinogen treatment to induce tumor growth. The resulting cMDI differ histopathologically from sMDI tumors. Whereas sMDI comprise adenocarcinomas, lymphomas, or histiocytic sarcoma/histiocyte-associated lymphomas [1], the predominant tumor entities of cMDIs were sarcomas. In addition, one spinocellular carcinoma model could be established.

These new models increase the number of available syngeneic tumor models. However, one cMDI model, JA-2017, did grow in immunodeficient SCID/bg mice only. Therefore, it seems to be not a “true” CBA/J H-2^k^ model. However, histologic characterization indicates that both primary and secondary tumors are of the same origin, since it displays an identical diagnosis of well to moderately differentiated anaplastic sarcoma. Hence, the most suitable explanation for growth restricted to immunocompromised SCID mice might be because that JA-2017 tumor is highly immunogenic, resulting in a rejection of tissue transplanted into syngeneic immunocompetent mice. This should be verified, e.g., by specific immunosuppression in immunocompetent mice. It is well known that carcinogen-induced tumors are often more immunogenic than spontaneous tumors [10], mostly caused by the expression of TATA [5,6,7]. Thus, the occurrence of just one tumor with high immunogenicity in the present study was rather surprising. In other cases, e.g., of MCA-induced fibrosarcomas in rats, 75% of the animals displayed TATA antigen expression and showed stronger immunogenicity [8]. In the case of chemically induced lung tumorigenesis and antigen expression, though, strain-specific differences were found with high tumor susceptibility in A/J mice and with intermediate susceptibility in BALB/c mice, whereas C57BL/6N or DBA/2N mice resisted carcinogen treatment (reviewed in [3]). For various immunotherapeutic approaches, it would be very interesting to identify if TATA antigens were present in cMDI models [11]. The observed striking invasion by CD4^+^ and/or CD8^+^ lymphocytes in 3 of 4 CBA/J cMDI models (except for JA-2011) seems to be a first indicator of such an enhanced immunogenicity and the presence of TATA in carcinogen-induced tumors (Figure 4B). In this context, it must be clarified why this does not lead to immediate tumor rejection (Figure 3) or to higher sensitivity to anti-ICPI antibody immunotherapy (Figure 5A–D). The overall effects of the anti-mPD-1 or anti-mCTLA-4 antibody monotherapies could be divided into “non-responder” (JA-2011), “moderate responder” (JA-2041 and JA-2042), as well as “responder” (JA-2019) models. Except for JA-2011, the reactivity pattern was very similar (Figure 5). In “moderate responder” and “responder” models, anti-mPD-1 caused a homogeneous individual tumor growth reduction, whereas anti-mCTLA-4 nearly completely inhibited the growth of tumors in all three models, but in only about 60% to 70% of the mice (Figure 5B–D). The monotherapies led merely in JA-2019 (“responder”) to a general and significant tumor growth inhibition by each of the two antibodies (Figure 5B). Nevertheless, there was no explanation for the resistance of certain cMDI to anti-mCTLA-4 or anti-mPD-1 monotherapies either within all (JA-2011) or single individual mice (JA-2041, JA-2042) in the other models (Figure 5A,C,D). However, the rather unexpected complete inhibition of tumor growth in all mice, observed by combined treatment with both anti-mPD-1 and anti-mCTLA-4 antibodies in JA-2042 (Figure 5D), might be a first indication of how to overcome the problem. Results showed that JA-2042 cMDI tumors basically display sufficient immunogenicity to become completely inhibited. However, considering present studies, it remains unclear whether the varying effects in individual mice relate to different properties (e.g., antigenic pattern) of individual tumor pieces or mirror a varying immune status of individual syngeneic mice, and if they caused by a combination of both, or by chance, or an unknown mechanism. Thus, in future, the outcome of such a combinatorial effect must also be investigated in the other cMDI models.

The resistance of “non-responder” model JA-2011 to anti-ICPI treatment seems to display another phenomenon. JA-2011 showed the fastest and most homogeneous growth of the four investigated cMDI models. Compared to the other anti-mCTLA-4 sensitive models, JA-2011 displayed a different pattern of tumor-infiltrating leukocyte populations, with a low infiltration of lymphocytic CD4^+^ and CD8^+^ effector cells, and many neutrophils (Figure 4B). This could be the explanation for the bad prognosis and tumor-supporting conditions, as well as for the low or missing antibody response [2,12,13]. On the other hand, JA-2011 was the only MNU-induced tumor, whereas the other tumors were induced by MCA. Former studies have shown that different carcinogens could induce carcinogen-specific mutations, which might then differentially activate various oncogenes and affect progression to malignancy [14]. Additionally, various carcinogens may also differ in their ability to transform the one or the other cell type [15] and, as already mentioned above, strain-specific differences in carcinogen susceptibility [3] might also influence the experimental outcome. In contrast, the data do not provide any evidence what might cause the resistance to antibody treatment e.g., in ICPI non-responder JA-2011.

The first RNA sequencing data regarding the expression of tyrosine kinase receptors, IFN-γ-signature or immune cell population markers, showed various patterns comparing the “non-responder” JA-2011 and “moderate responder” JA-2042 models (Appendix A [1]). The relationship of expression patterns between individual models demonstrates very different gene expression patterns between two related cMDI sarcomas (JA-2011 and JA-2042). It remains unclear if the differences might be caused by different routes (s.c. versus i.m.) or carcinogens (MNU versus MCA) used for induction, and if they are also related to the different response patterns. Flow cytometry confirmed that enhanced Cd4 gene expression in JA-2042 was actually associated with enhanced CD4^+^ T cell infiltration by various tumor-infiltrating leukocytes. To assess this question, future studies need to be performed to characterize the gene expression of further gene markers, such as immune cell-specific genes [2] or tumor-specific genes [16], and compare these with the gene expression in respective primary human samples [17]. In addition, this analysis should be extended to further established sMDI and cMDI tumors.

The establishment of the MDI models opens a door for the better understanding of tumor antigenicity, conditions, potential signals, and the role of tumor infiltration by immune cells and their relation to response to therapeutic interventions. The various outcomes and questions regarding MDI models of, e.g., possible tumor subgroups, histological heterogeneity, individual TATA, and varying sensitivity to treatment, remains unresolved, since this was not the matter of the exemplary present investigations. In the present two papers, we introduced new mouse-derived isograft models and described their establishment, characteristics, and potential for future research.

It can be assumed that the MDI model variety will be further enhanced by establishing cMDI and sMDI in additional inbred mouse strains with immunological differences [18,19,20,21], but also genetically heterogeneous in angiogenesis [22], with various susceptibility to drugs [23], or with varied strain-specific susceptibility for metastasis [24].

To sum up, we can state that MDI models are promising and effective in vivo cancer research tools. They reflect the clinical situation better than other in vivo mouse tumor models. cMDI and sMDI models allow for comparison of new therapeutic concepts against either specific tumor entities or different MHC backgrounds. Thus, the combination of the two model types will be more suitable and relevant for studying the interactions between stroma, immune cells, and tumor environment in cancer progression, metastasis, and therapy in a completely natural host.

## 4. Materials and Methods

### 4.1. Mouse Strains

All in vivo experiments were performed in accordance with the German Animal License Regulations (Tierschutzgesetz), identical to UKCCCR Guidelines for the welfare of animals in experimental neoplasia (license: G-15/160; Regierungspräsidium Freiburg) [25]. CBA/J mice at 12 weeks of age from Charles River Laboratories, Sulzfeld, Germany were used for all studies. Mice were housed under pathogen-free conditions in conventional cages stored in Scantainer ventilated cabinets, Scanbur, Denmark, with autoclaved nesting material, and cardboard tunnels. Mice were housed on a 12/12 h light/dark cycle, with ad libitum autoclaved water and M-Zucht rodent diet, ssniff Spezialdiäten GmbH, Soest, Germany. Four animals were housed in one cage; however, if any animals exceeded 30 g body weight during the observation period, just 2–3 animals were housed within one cage.

### 4.2. Monitoring of Animals

Animal weights were measured twice weekly during observations and three times weekly during the treatment studies (balance: Mettler Toledo PB602-L). Animal behavior was monitored daily.

Termination criteria: If the volume of tumors exceeded 2000 mm³ or an edge length of 2 cm, tumor ulcerations, weight loss of >20%, cachectic phenotype (tumor cachexia), remarkably abnormal, non-physiological posture as a sign of pain, apathy (severe inactivity), strongly reduced feed and water intake, severe dyspnea, motor deficit manifestations or paralysis, ascites, persisting diarrhea, massive behavioral changes, or other unexpected signs were observed, indicating an obviously heavy burden of the animals, the mice would be immediately sacrificed, according to the ATBW/GV-SOLAS.

### 4.3. Carcinogen Induction of Tumor Growth

To establish primary cMDI tumors, animals were treated once subcutaneously, intravenously, intramuscularly, intraperitoneally, or 3 times orally with either 3-methylcholanthrene (MCA) or *N*-methyl-*N*-nitrosourea (MNU) (Sigma-Aldrich Chemie GmbH, Steinheim, Germany) followed by short-term (3–9-month) monitoring for tumor appearance. The animals were treated according to the following schemes (Table 1): Group one—four female mice were treated subcutaneously on the right and left flank (s.c.) and once additionally into one intramammary fat pad (imfp) with 50 mg/kg MNU per injection in sterile 0.85% NaCl (pH 5.0). Four male mice of this group were also injected s.c. and once intravenously (i.v.); Group two—four male mice (3 animals died within 2 days after injection for an unknown reason) were treated once intraperitoneally (i.p.) with 100 mg/kg MNU in DMSO (Sigma-Aldrich Chemie GmbH, Steinheim, Germany) [26,27,28,29]; Group three—four male and female mice each were treated once s.c. with 0.1 mL (5 mg/mL) MCA in sesame oil (Sigma-Aldrich Chemie GmbH, Steinheim, Germany) and once additionally with 0.025 mL intramuscularly (i.m.) on the right hind leg [30,31]; Group four—four male and female mice each were orally (p.o.) treated 3 times in weekly intervals with 0.3 mL of an MCA emulsion (5 mg/mL) in sesame oil [32].

When animals met termination criteria, the handling of tumors or any suspicious tissue was done in the same way as described already for spontaneous ones [1]. Briefly, tumors and suspicious tissues were assessed macroscopically and cut into small pieces (of 2 to 3 mm^3^) in sterile PBS using a scalpel. The tumor pieces were either stored in 10% DMSO freezing medium at ≤−80 °C or directly implanted s.c. into 5- to 6-week-old sex-matched syngeneic and/or female immunodeficient SCID/bg mice using a trocar. Two hours prior to transplantation, mice received the analgesic Meloxicam (Metacam^®^; 1 mg/kg, Boehringer Ingelheim Vetmedica GmbH, Ingelheim, Germany) s.c. Implantation was performed under anesthesia with 2 to 3 volume percent isoflurane in combination with an oxygen flow rate of 0.6 L/min. Primary tumor volumes were determined by caliper measurement two times a week. Tumor sizes were calculated according to the formula W^2^ × L/2 (L = length and W = the perpendicular width of the tumor, L > W).

### 4.4. Mouse-Derived Isograft (cMDI) Re-Transplantation

Frozen tumor pieces were thawed rapidly at 37 °C and washed twice in ice-cold PBS, and then implanted as described above. The growth of transplanted tumor pieces was observed until termination criteria were met. During necropsy, tumor tissue but also other suspicious tissues were collected (F1). To amplify the number of tumor pieces derived from one primary tumor, tumor pieces were generated and re-transplanted again into mice (F2). Excess of tumor pieces were frozen stored at ≥−80 °C or in the vapor phase of liquid nitrogen.

Amplification and sample collection were repeated several times, and samples were called F1–Fn (*n* = F3–6 amplifications) [33]. Some tumor pieces (primary and re-transplanted) were fixed in formalin for further analysis. The model development was finished by successful re-transplanting and testing the outgrowth of frozen tumor pieces into 4 to 6 syngeneic animals.

### 4.5. cMDI—Histological and Pathological Analysis

Wet tissues were collected in embedding cassettes and formalin-fixed in 4% neutral buffered formaldehyde (Engelbrecht, Edermünde, Germany) at room temperature for about 24 h, followed by automatic dehydration and embedding in IHC-grade paraffin using a Leica TP 1020 (Leica Biosystems, Nussloch, Germany) and Leica Histo Core Arcadia H/C (Leica Biosystems). FFPE tissue blocks were stored at room temperature. Sections from paraffin blocks with non-specified tissues were cut into slices of approximately 2–3 µm thickness and routinely stained with HE at TPL Path Labs (Freiburg, Germany).

Histopathological examinations were then performed blind using a Zeiss Axioscope microscope (Carl Zeiss, Jena, Germany) at a magnification of up to 400×, by one of the authors (T.L.). Digital microphotographs were taken using a Nikon Digital Sight DS-Fi camera (Nikon Instruments Europe B.V., Amsterdam, Netherlands). Whole slide imaging using an Axioscan Z1 (Carl Zeiss) were also performed from all case, for the purpose of comprehensive iconography.

### 4.6. cMDI—Establishment and Efficacy Studies

To assess growth curves as well as take rates of cMDI, frozen tumor pieces were transplanted into 12 sex-matched animals. In the following, animals were monitored as described above. Deviation of the health status was documented, and animals were euthanized individually before study termination when ethical termination criteria were met. Take rate was defined as the percentage of total tumor-bearing versus all animals during the observation period. From primary re-transplanting studies, it is known that in some cases, animals could also develop tumors later, but this growth was not included in the analysis. *Real running time* ascertains the actual number of live tumor-bearing animals at study end. This number of animals may be smaller than defined by take rates, since animals could be lost by fast tumor growth (i.e., termination by tumor volume), were repeatedly found dead for unknown reasons, or were lost by unexpected ulcerations (termination for ethical reasons) before the end of the observation period. Determination of *appearance* denotes the time range at which randomization of a sufficiently large number of animals with tumor growth is possible, and whose tumor properties (e.g., volumes, growth rate) must be appropriately taken into consideration for randomization criteria. For randomization, a robust automated random number generation within individual blocks was used (MS-Excel 2016, Microsoft, Redmond, WA 98052, USA). Both *appearance* and *real running time* allow for defining how many animals must be implanted with tumor pieces to randomize about 10–12 animals per group at *appearance*, which will also still be alive at the end of the observation period.

Treatment was initiated on the day of randomization, according to the following schedules: vehicle control (PBS) and anti-ICPI antibodies (anti-mPD-1, anti-mCTLA-4, or combination of anti-mPD-1/anti-mCTLA-4) were injected i.p. 3 times every third or fourth day (Q3d3/4x), with a dosage of 10 mg/kg and an injection volume of 10 mL/kg (combination 5 mL/kg). Tumor growth and effects regarding tumor growth reduction due to therapeutic intervention were followed by caliper measurement until the end of observation. The study was finalized by necropsy, tumors were resected, and tumor volumes and wet weights in the experimental groups were determined and documented. In some cases, parts of tumors from various experimental groups were also formalin-fixed and paraffin-embedded, and used for flow cytometry analysis and/or RNA sequencing. Probability (*P*) was tested with a parametric unpaired *t* test (GraphPad Prism 5.04, GraphPad Software, San Diego, CA 92108, USA) compared to PBS vehicle control. Differences were determined as not significant with ns > 0.050 and as significant * with *p* < 0.050, ** with *p* < 0.010, and *** with *p* < 0.001.

### 4.7. Flow Cytometric Analysis of cMDI Tumor-Infiltrating Leukocytes

Primary tumor material (at least approx. 300 mg) was disrupted using gentleMACS™ C Tubes (Miltenyi Biotec, Bergisch Gladbach, Germany) containing the enzyme mix of the Tumor Dissociation Kit (mouse) according to the manufacturer’s instructions (Miltenyi Biotec). Following that, erythrocytes were removed with the Red Blood Cell Lysis Solution (Miltenyi Biotec). The obtained single cell suspensions were counted and dispensed in two 96-well plates, if possible, at 3 × 10^6^ cells/well (if there were, in total, less than 2 × 10^6^ single cells in a tumor sample, the T cell panel was preferred for staining). The single cells were washed with PBS (Capricorn, Ebsdorfergrund, Germany) and stained for living cells using eBioscience™ Fixable Viability Dye (Thermo Fisher, Nidderau, Germany). After washing and centrifugation (400 *g*), the samples were incubated with 50 µL/well Fc block (anti-mouse CD16/CD32, 1:50) for 30 min in FACS buffer. Thereafter, a 2× concentrated master antibody mix (T cell panel with antibodies against murine CD45, CD3, CD4, CD8a, CD25, CD44, myeloid cell panel with antibodies against murine CD45, CD11b, CD11c, Ly6G, and Ly6C, and macrophage panel CD45, CD11b, F4/80, MHC II, CD206) was added to each well (50 µL) and incubated for 30 min in the dark. After washing, cells stained for the myeloid or macrophage cell panel were fixed. Cells stained for the T cell panel were prepared for intracellular staining. Intracellular staining was primed by adding 50 µL fix/perm (Thermo Fisher Scientific) buffer for 30 min. Thereafter, 1× permeabilization buffer was added and centrifuged at 836 *g*. The cell pellet was resuspended in 1× permeabilization buffer containing the anti-FoxP3 antibody and incubated for 30 min in the dark. After two times washing with 1× permeabilization buffer, cells were washed with FACS^TM^ buffer. The cells were kept at 4 °C in the dark until analysis, no later than 5 days after preparation. The samples were analyzed by flow cytometry using an BD LSR Fortessa^TM^ (Becton Dickinson, Heidelberg, Germany). All antibodies were purchased from Thermo Fisher except for CD3 and CD206 (BioLegend, San Diego, CA 92121, USA), and CD8a (Becton Dickinson).

### 4.8. cMDI—RNA-Seq

For details on RNA-Seq whole transcriptome shotgun sequencing analysis, please see the accompanying paper [1].

## 5. Conclusions

The established MDI models described in the two related papers are a new type of promising in vivo cancer testing tools representing primary spontaneous [1] or carcinogen-induced tumors of a low passage number that are propagated in mice only, without any tissue culturing. Therefore, the models possess conserved tumor characteristics and intratumoral immune cell populations, which were histologically and genetically characterized. Carcinogen-induced tumors established in this way represent outgrowing tumors (or metastases) which have overcome the body´s own regulatory mechanisms, and are thus comparable to clinically manifest tumors, without any previous or further in vivo or in vitro manipulation. Since chemical induction resulted mostly in sarcomas, therapeutic effects in various tumors of similar origin can be investigated. The first results indicated that individual tumors—despite their relatedness—might differ in their therapeutic sensitivity.

Future inclusion of other mouse strains with different MHC classes and immunological backgrounds will enhance the therapeutic window. If not be restricted to only one MHC class and immunological archetype, the models will correspond to experiments in various “individuals” but with the possibilities to repeat or vary treatment concepts. Additionally, further refinement of the established models will be possible. Use of relapsing tumors from original ones after various kinds of treatments (immunological, chemotherapeutic, irradiation, or combinations thereof) also open possibilities to search and find follow-up concepts for re-treating therapy-resistant relapsing tumors.

Together, this opens a broader field of experimental in vivo cancer therapeutic concepts.

## Figures and Tables

**Figure 1 cancers-11-00242-f001:**
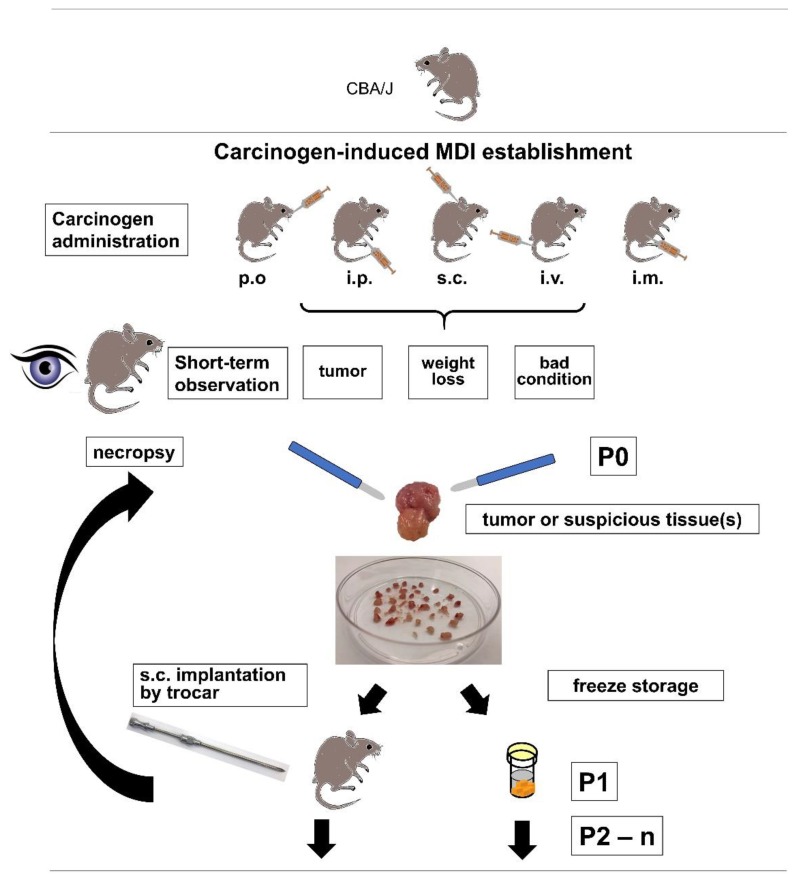
Scheme of cMDI establishment. Various carcinogen administration routes, as well as termination criteria, and general handling of tumor or other suspicious tissue samples are schematically summarized (P: passage numbers; P0: primary necropsy; P1: first passage of primary tumor; P2–n: second and further follow up passages).

**Figure 2 cancers-11-00242-f002:**
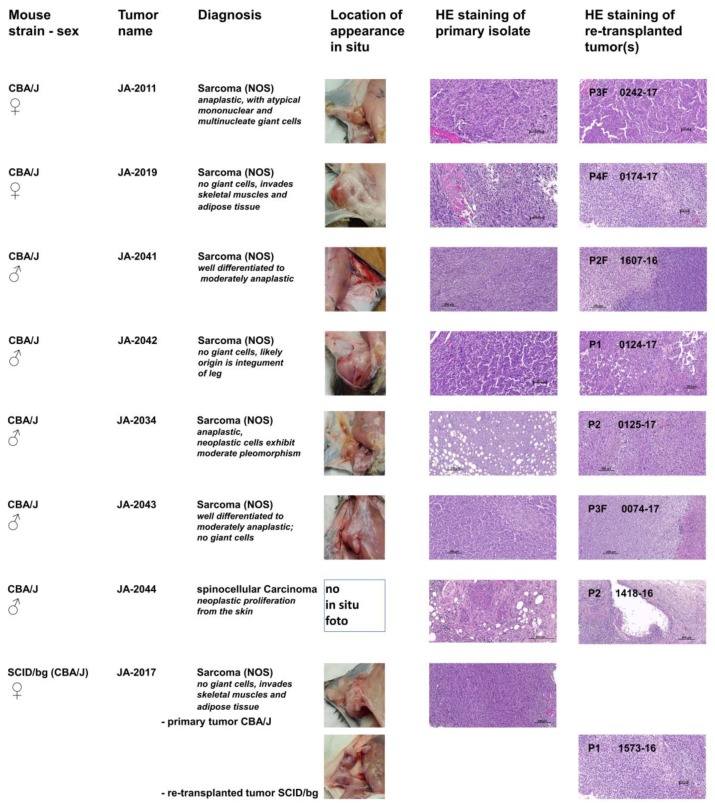
Histopathological characterization of established cMDI tumors. In situ location of primary tumor or suspicious tissue during necropsy, as well as hematoxylin and eosin (HE)-staining of resected primary and re-transplanted tumors using directly isolated (of passages P1–P4) or frozen tumor pieces (of passages P2F–P4F) are documented for eight cMDI models. NOS—sarcomas could not be otherwise specified by HE-staining. JA-2017 originated from female CBA/J but only grew in immune-deficient SCID/bg.

**Figure 3 cancers-11-00242-f003:**
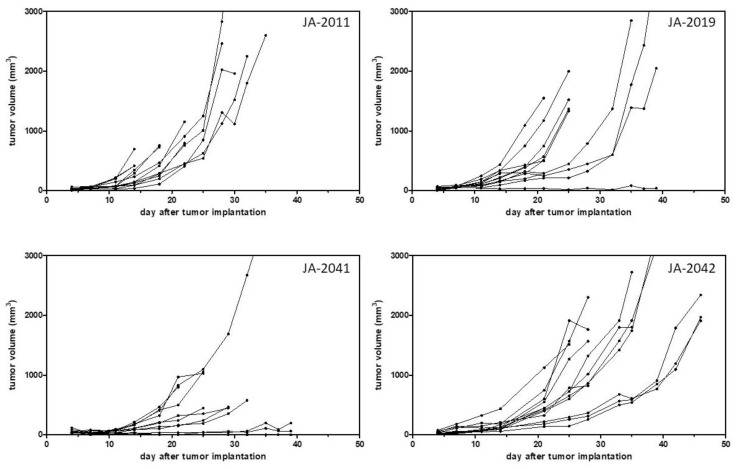
Take rates and growth curves of four cMDI models. Growth curves of 10–12 mice each implanted with frozen tumor pieces are shown. Tumor growth was measured twice weekly by calipering.

**Figure 4 cancers-11-00242-f004:**
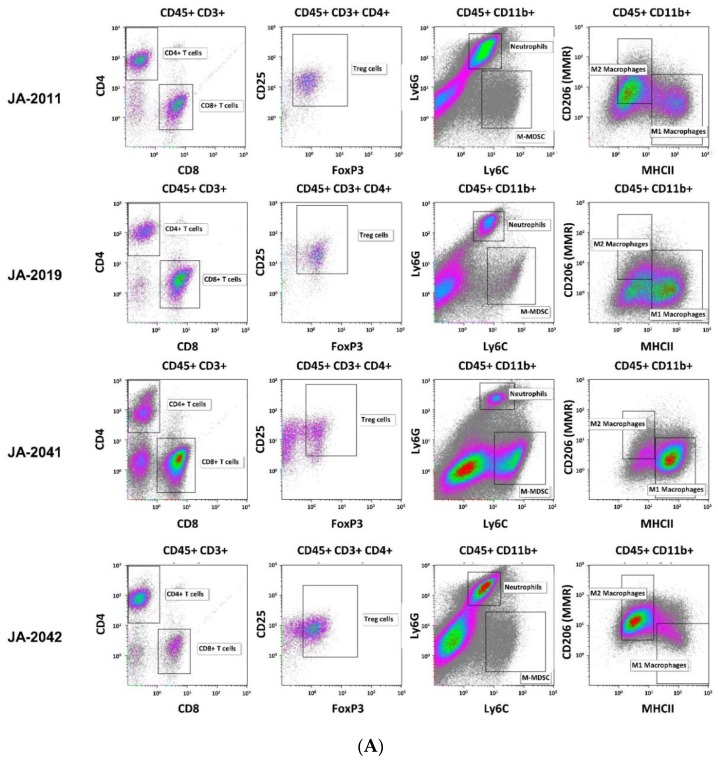
(**A**). Flow cytometric analysis of cMDI tumor-infiltrating leukocytes. Representative flow cytometry plots of tumor-infiltrating CD4^+^ and CD8^+^ T-cells, T reg cells, M-MDSCs, neutrophils/G-MDSC (cells of CD11b^+^, Ly6G^+^, Ly6C^int^ phenotype), and M1/M2 macrophage populations isolated from untreated syngeneic cMDI tumors JA-2011, JA-2019, JA-2041, and JA-2042. M-MDSC = monocytic myeloid-derived suppressor cells, MMR = macrophage mannose receptor, Treg = regulatory T-cells. (**B**). Quantitative flow cytometric analysis of cMDI tumor-infiltrating leukocytes. Flow cytometry analysis of tumor-infiltrating immune cells (quantitative analysis) in untreated syngeneic cMDI sarcoma JA-2011, JA-2019, JA-2041, and JA-2042 models and seven established syngeneic cell line-based mouse models of CD4^+^/CD8^+^ T-cells, M-MDSCs, neutrophils, M1/M2 macrophage, and Treg cell populations isolated from untreated tumor tissue. The graphs depict the number of cells of each population per 1 × 10^6^ living leukocytes. For further information regarding depicted syngeneic established cell line-derived standard tumors and experimental details, see Appendix A [1].

**Figure 5 cancers-11-00242-f005:**
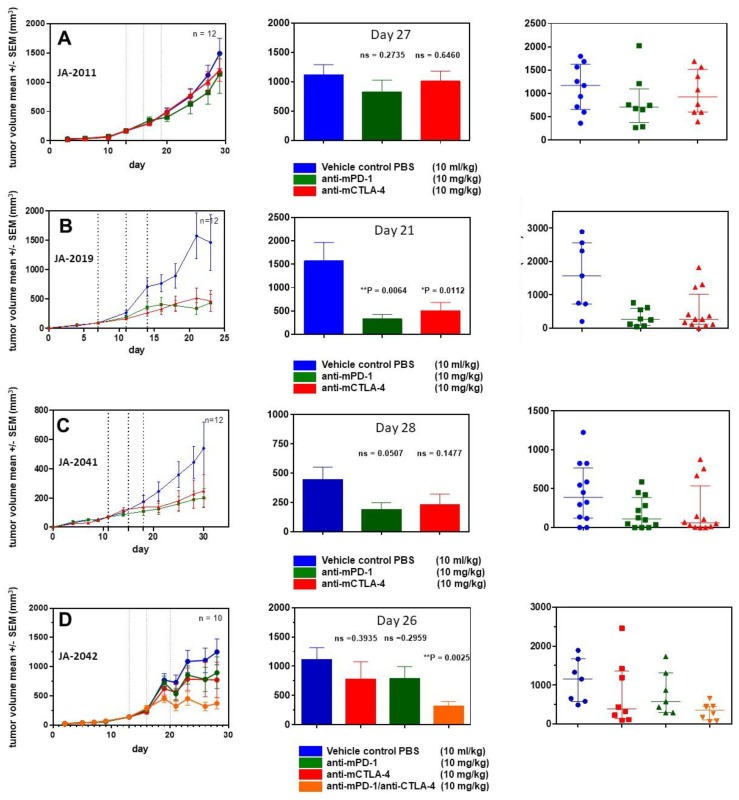
Efficacy studies of selected cMDI models using immune checkpoint inhibitor antibodies. Efficacy of immune checkpoint inhibitor antibodies (anti-PD-1, anti-CTLA-4, or combinations thereof) was investigated in four selected cMDI models (**A**) JA-2011, (**B**) JA-2019, (**C**) JA-2041, and (**D**) JA-2042. Ten to 12 sex-matched CBA/J mice per group were treated three times (details of treatment schedule—see Material and Methods) with antibodies or PBS as vehicle control (dotted lines). Results are shown as mean (curve and bar graphs) and individual values of single mice (dot plots) with standard error of the mean (SEM). Probability (*p*) was tested with a parametric unpaired *t* test (GraphPad Prism 5.04) as compared to PBS vehicle control. Differences were determined as not significant with ns > 0.050 and significant * with *p* < 0.050, ** with *p* < 0.010, or *** with *p* < 0.001.

**Table 1 cancers-11-00242-t001:** Carcinogen application schedules for the induction of carcinogen-induced mouse-derived isograft (cMDI) models.

	MNU		MCA	
	**♀ CBA/J Mice**			
**Application Route**	s.c./imfp		s.c./i.m.	p.o.
**Animal Number**	4		4	4
	**♂ CBA/J Mice**			
**Application Route**	s.c./i.v.	i.p.	s.c./i.m.	p.o.
**Animal Number**	4	1 (3) *	4	4

CBA/J mice of both sexes were treated with either 3-methylcholanthrene (MCA) or *N*-methyl-*N*-nitrosourea (MNU) via different application routes (s.c. subcutaneous, imfp intramammary fat pad, i.m. intramuscular, p.o. per oral, i.v. intravenous, i.p. intraperitoneally). * (3/4) animals died within 1–2 days after injection for unknown reasons.

**Table 2 cancers-11-00242-t002:** Summary of established cMDI models.

Carcinogen-Induced MDI								
**Tumor Name**	**JA-2044**	**JA-2011**	**JA-2019**	**JA-2034**	**JA-2041**	**JA-2042**	**JA-2043**	**JA-2017**
**Hematopoietic**	no	no	no	no	no	no	no	no
**Histopathological Diagnosis**	spinocellular carcinoma	Sarcoma NOS, anaplastic,	Sarcoma NOS well-differentiated	Sarcoma NOS, anaplastic	Sarcoma NOS well-differentiated to moderately anaplastic	Sarcoma NOS well-differentiated to moderately anaplastic	Sarcoma NOS well-differentiated to moderately anaplastic	Sarcoma NOS, anaplastic,
**Mouse Strain**	CBA/J	CBA/J	CBA/J	CBA/J	CBA/J	CBA/J	CBA/J	CBA/J **only in SCID/bg**
**Sex**	**♂**	**♀**	**♀**	**♂**	**♂**	**♂**	**♂**	**♀**
**Carcinogen** **(Appl. Route)**	**MCA** **(s.c.)**	**MNU** **(s.c.)**	**MCA** **(s.c.)**	**MNU** **(s.c.)**	**MCA** **(s.c.)**	**MCA** **(i.m.)**	**MCA** **(s.c.)**	**MCA** **(s.c.)**
**Estimated Growth Time (in Days)**	39–113	14–35	16–63	29–55	25–105	25–46	17–36	14–28
**Included Animals**	frozen/directly (2/6) *	growth curve (12) **	growth curve (12)	frozen/directly (2/5)	growth curve (12)	growth curve (12)	frozen/directly (4/5)	frozen/directly (2/3) only in SCID/bg
**Take Rate in % Number of Animals**	63% 5/8	100% 12/12	100% 12/12	57% 4/7	83% 10/12	100% 12/12	100% 9/9	100% 5/5
***Appearance* (*AP*) on Days**	nd	10–12	4–7	nd	7–11	9–13	nd	nd
***Real Running Time* (*RRT*) on Days** **(% Alive Animals)**	nd	22 (87%) 30 (54%)	21 (75%) 25 (54%)	nd	25 (95%) 30 (64%)	28 (97%) 39 (64%)	nd	nd

The table summarizes the characteristics of cMDI tumors. In the case of JA-2017, re-transplanted growth was observed only in SCID/bg but not in syngeneic female mice. Estimated growth times in established models are the mean or range of tumor growth until termination of at least 12 tumor-bearing animals (growth curves) **. In the other established models, growth times were calculated from the outgrowth of frozen or directly * re-transplanted tumor pieces in various experiments. Take rates reflect the percentage of animals developing a tumor within an estimated growth time. Appearance is defined as the earliest time point(s) allowing robust randomization at mean tumor volumes between 40 and 150 mm^3^ in re-transplanted animals. Intense variation in animal numbers is mostly caused by unexpected death or termination due to ethical reasons. Thus, we defined the so-called real running time (RRT) of the models. RRT determines the time difference from implantation to the time point when the remaining animal number reaches ~60% of the starting animal group size. Thus, it also defines TTW, i.e., the maximal treatment time window to treat animals from randomization (AP) to the potential study end. This allows for calculation of a realistic study length, as well as necessary group size for statistically reliable analysis of, e.g., immune checkpoint inhibitors. Included animals: * resulting from frozen or directly re-transplanted tumor pieces (number of animals), ** growth curves resulting from re-transplantation of frozen tumor pieces into 12 animals each. Bold—growth in another strain only; Grey columns are not yet characterized by growth curves (below-4) or the tumor grows only in another mouse strain (above-1)

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
