# Peer review of "Mouse-Derived Isograft (MDI) In Vivo Tumor Models II. Carcinogen-Induced cMDI Models: Characterization and Cancer Therapeutic Approaches"

_cancers, 2019, doi:10.3390/cancers11020242_

Reviewer 1 Report

In this study, Beshay et al established mouse-derived isograft in vivo tumor models by using carcinogen induction, and characterized these so-called cMDI models. Their findings show that carcinogen-induced cMDI models have more efficiency in tumor induction but the resulting induced tumors are mostly sarcomas. In addition, by using flow cytometric analysis, The cMDI model can provide in vivo tumor tissue and intratumoral immune cell populations that exhibit highly conserved tumor characteristics. In general, the present study establishes in vivo isograft tumor models and well analyzes the characteristics of the cMDI models. The study design is satisfactory, the experiments have been well performed, and the characteristics of the tumor models are clearly demonstrated. Although cMDI is limited to sarcoma, it suggests that these cMDI tumor models should be useful and useful for cancer treatment research.

Author Response

cMDI - Author’s Reply to the Review Report - Reviewer1

Dear Madam, dear Sir,

Thank you very much for critical reviewing and your helpful suggestions for revision of our manuscript cancers-409369 “Mouse-Derived Isograft (MDI) in vivo Tumor Models II. Carcinogen-Induced cMDI Models: Characterization and Cancer Therapeutic Approaches”.
According to criticism regarding excessive switching between tumor entities in the manuscript by other reviewers, we restructured the presentation of individual tumor entities, and also enhanced cross reference to the presented supplementary data in a more precise manner (see revised manuscript).
We hope revised manuscript will possibly improve presentation of the manuscript and find again your acceptance. Thank you for your support.

Kind regards,
Peter

(on behalf of all coauthors)

Reviewer 2 Report

In a second study to their description of the “Mouse-Derived Isografts (MDI) from spontaneous tumors (sMDI), Beshay et al.,  now describe results from an analysis of syngeneic in vivo models derived from carcinogen-induced mouse derived isografts (cMDI).   In general, in contrast to the sMDI, cMDI results in a greater number of “individual, but histologically related tumors” particularly of the sarcoma phenotype.  There are several questions which emerge from reading through this study”

General Concerns:

In general, the Tables and Figures are hard to understand.  There are missing details when comparing Table 1 with Figure 1.   Ex  Fig 1.  What happened to mice given a “i.p” route of carcinogen administration for the female mice?

Section 2.2 and Figure 2.  While there are terms such as  “anaplastic” or “differentiated” or “giant” cells” mentioned in Figure 2 (under Diagnosis) there is no indication of these features on the figures of the primary, or re-transplanted tumors.   Specific details about how these tumors differ are not clear at all from the very small figures.    For example, where is the nerve bundle mentioned on page 5, line 137?

And there are other missing features, such as “deeply invasive”.  How is that seen in the figures?.

I do not understand the difference in tumor growth variability between tumors characterized in Fig. 3, and the same tumors used in Fig. 5 for testing effects of checkpoint inhibitors.  For example, tumor JA-2019 has an extremely variable growth rate between different mice, and that is interesting, demonstrating heterogeneity.  Yet, this same tumor, shown in Fig 5 B shows a very uniform control tumor growth rate with little variability (compare for example, day 15 in both figures.)    This same concern exists for Tumor JA-2042.  How do the authors know that at least some of the effects of anti-checkpoint inhibitors in those groups are not also due to having randomly included slower growing tumors in those groups (i.e., as seen with NO treatment in Fig. 3)?

Page 5, line 132 indicates that in some cases, mitotic index was performed, as well as inflammation.  Where is that data?  

RNA sequencing is mentioned in several places, (in one example, page 15, line 473 and page 16, lines 504)  but no data is provided (?).

While a lot of flow data on immune infiltrates is given, and compared to other cell-line derived tumors, no comparison, or conclusions related to tumor growth rate differences, are given (except for JA11 in the Discussion).    What was the point of comparing the cell lines? This should be clarified more completely in the paper. 

Other points:

Table 1:  Why are symbols for male and female presented as double symbols?  Are there hybrids being used? 

Many terms are unusual: Abortion?  “conspicuous” tissues?  What is conspicuous about these tissues??

Why is there a “(16)”  after the 13 in line 75, page 2?   If this is because of the addition of three mice that later died,  this should be indicated.

i.mfp  should be simply imfp.

Necropsy is misspelled in Fig. 1

Author Response

sMDI/ cMDI -Author’s Reply to the Review Report - Reviewer4 sMDI and Reviewer2 cMDI

Dear Madam, dear Sir,

Thank you very much for your detailed and critical reviewing as well as your very helpful comments and suggestions for revision of our manuscripts cancers-409311 “Mouse-Derived Isograft (MDI) in vivo Tumor Models I. Spontaneous sMDI Models: Characterization and Cancer Therapeutic Approaches” and cancers-409369 “Mouse-Derived Isograft (MDI) in vivo Tumor Models II. Carcinogen-Induced cMDI Models: Characterization and Cancer Therapeutic Approaches”.
Considering your remarks we tried to remove the substantial weaknesses by extensive revision of our two manuscripts. In the following, I would like to address some of the major topics of your valid and substantial critics. The point to point answers are given in the order concerning manuscript 1#, 2# or both ones.

(manuscript #1)
- wrong arrangement of Figs. 1b and 1a was the result of editorial processing of the manuscript #1. We had overlooked that reference to Fig. 1b preceded reference to Fig. 1a in the manuscript text, and thus the editor sorted the figures in reverse direction. They are now renamed, and in chronological order.
- The descriptions between 2.1 establishment and 2.2 Histology/pathology are repetitious, and overall very confusing: Familiar with the about 3-year long history of establishing the MDI models, our writing and reading of the manuscript(s) was “biased” by the logic of each step gone on this way. To introduce into the strategy and logic of our doing we thought -and I still do so- that a description of MDI establishing along the timeline (like a history) might be helpful to better understand the steps on this way.
Therefore, in chapter 2.1 we provide now a “historical” overview of establishing process while clearly separated chapter 2.2 shows histopathological diagnosis of the various sMDI tumor either of hematopoietic or non-hematopoietic origin.
- Origin of the tumors – tumor type of these models: As accounted now in the text, in most cases, we could not determine the origin of primary tumor/suspicious tissue isolates. This means that we could actually not differentiate if isolates represent the primary tumor or rather secondary metastatic tissues. Tumorous tissue isolated from the lung of JA-0018 CBA/J (lymphoma) or enlarged axillary lymph nodes found in JA-0009 DBA/2N mice (adenocarcinoma) exemplary illustrate the problem. Thus, we would like definitely to point out that in case of sMDI (in cMDI with primary tumors on carcinogen injection sides the situation is another one) the preliminary organ/tissue origin of the tumors only could be verified by histopathological findings.
- sMDI JA-0021: In case JA-0021 the Origin of the tumor also remained unclear for a time. But here, the exclusive growth of the tumor only in SCID/bg mice additionally raised the question in which mouse strain this sMDI originated. The histological finding that primary C3H/HeJ isolated lymph node showed a questionable lymphoma or even non-malignant morphology indicates that malignant tissue might probably originate also from the suspicious, re-
transplanted SCID/bg mouse thymus tissue. This is communicated now more distinct in the text, but exact origin has finally to be clarified.

(manuscript #2)
- regarding “i.p.” route of carcinogen administration in female mice: We started i.p. application only in male animals surprisingly and alarmingly 3/4 animals died within 1-2 days. The fourth one, however, was well and did not show any impairments. Since we really did and still do not have any idea why the animals died, we did not treat the female mice via the i.p. route, i.e. this group was removed from the study.
- I do not understand the difference in tumor growth variability / tumor JA-2019 has an extremely variable growth rate in Fig. 3 - in Fig 5 B shows a very uniform control tumor growth rate: if you compare the data, the mentioned discrepancy between tumor growth variability in figs. 3 and 5 is probably only a perceived one. Whereas in Fig. 3 individual growth curves with different growth length of single mice are shown, Fig. 5 shows the mean curve of the vehicle group with respective SEM values during the treatment period. However, the actual growth heterogeneity in the vehicle group of efficacy study (Fig. 5), is better to realize considering individual tumor volumes at day 21 shown in the right dot plots (ranging from about 150 – 2,800 mm3). A similar picture was seen already on day 15 (not shown) with individual tumor volumes ranging from about 75 – 1,450 mm3.
- How do the authors know that at least some of the effects of anti-checkpoint inhibitors are not due to having randomly included slower growing tumors: We used for randomization 36 animals divided into three groups of 12 animals each, a robust automated random number generation within individual blocks (MS-Excel 2016), which guarantees a more or less, but statistically homogeneous distribution of differently growing tumors in the various groups (Materials and Methods). The necessary group size of 10 – 12 animals was calculated by neutral external statistician based on the growth curves shown in Fig. 3.
- no comparison, or conclusions related to tumor growth rate differences, are given (except for JA-2011 in the Discussion) - What was the point of comparing the cell lines?: At this time point of model characterization broadly varying individual growth time periods were observed in the four cMDI (Tab. 2). In contrast, their real running times (RRT), determining the actual treatment time window (TTW) as well as potential study endpoints are rather similar in untreated mice (Tab. 2, Fig. 5). Since these terms were not defined distinct enough, we changed their presentation and hope they are in an intelligible form now (chapter 2.3).
Since cMDI RRT were also very similar compared to growth curves (RRT) of “other cell line-derived” tumors (Fig. S1– sMD) we looked for common properties of the different models. However, in the seven cell line-based models, neither growth rate (varying from 18 – 25 days), nor hetero- (CT26.wt, Clone M3, or B16.F10) or homogeneity (LL/2, 4T1, RENCA; or MC38-CEA) of growth displayed any relation to the therapeutic outcome with immune checkpoint inhibitors (Fig. S1-sMDI).
Main aspect of comparing the four cMDI (but also sMDI) with cell line-based models, was the idea to compare the new MDI models with as many parameters as possible of the already often used and well characterized cell line models. Thus, we did not consider a relation between “fast and homogeneous growth” and “non-responder” status in cMDI. JA-2011 showed fastest and most homogeneous growth of the four investigated cMDI models. But the resistance of non-responder model JA-2011 to anti-ICPI treatment seems to display another phenomenon.

(manuscripts #1 and #2)
- “All abbreviations in figures need to be explained in figure legends.”: There is an entire abbreviation list at the end of the manuscript, and to my opinion a separate list in each figure would needlessly expand length of the legends. Additionally, each abbreviation in the text is explained with first appearance.
- “established” models / “finalization” / “models not finished yet” / “a paper with models that aren´t finished”: as already mentioned above “writing and reading of the manuscript(s) was “biased” by the logic of each step gone on this way ….”
We now changed and “homogenized” our terminology in the revised manuscripts to exactly describe the status quo of the respective models,
using the following terminology:
term “established” – now defined as: stable outgrowth from frozen tumor pieces, i.e. re-transplantable samples are permanently secured (i.e. not dependent on persistent in vivo amplification).
term “finalized” – now removed: models status quo is described, either i) histopathologically analyzed, ii) with determined growth curves, or iii) further characterized by flow cytometry and/or RNAseq analysis.
term “not (yet) finished” – now removed: these isolates, only primarily re-transplanted and outgrown once or a few times were now only accounted as potential MDI samples in the text or legend of tables.
- the micrographs are the size of postage stamps and none of the features detailed in the text can be seen / and other missing features / pathologist should be named: We summarize the relevant histopathological data as origin of primary isolate, H&E staining of this tissue and of another, s.c. re-transplanted follow-up-sample of the same tumor, together with tumor name, mouse strain, sex, and the tumor diagnoses, in order to submit as much information as possible in one (or two) figures each. I think, in general the figures are very informative, but not unusable to give more detailed information. Since assessment was the sum of several pictures, it could not simply be condensed within one figure by the pathologist. The initials TL of him, Thomas Lemarchand, had been already displayed in “Author Contributions” section.
After intense internal discussion, we decided on the following solution: Small corrections were applied (e.g. removing the double sex symbols) to the figures in the manuscript to still show the general information in one or two figures in the articles. But to allow evaluation in depth of the pathohistological data, we submit largely expanded supplementary data for each
manuscript: high resolution images of different magnification with the relevant features highlighted in the text for the analyzed tumors.
- Section 2.4 with the flow data has several problems.
What the authors are calling neutrophils are likely gMDSC (CD11b+Ly6g+CD45+). This should be addressed and clarified.
Our statement: G-MDSC are characterized as CD11b+, Ly6Clow/neg and Ly6Ghigh phenotype, whereas for neutrophils a rather CD11b+, Ly6G+, Ly6Cint phenotype. Cells gated in our manuscripts as neutrophils, i.e. of CD11b+, Ly6G+, Ly6Cint phenotype, could not definitely be discriminated from G-MDSC of CD11b+, Ly6Clow/neg and Ly6Ghigh phenotype, i.e. both cell types could be contained in the gated population. Our gating was performed by using FMO controls, and we tend from our experience the population rather to be Ly6Cint then Ly6Clow/neg (Figs. 5A-sMDI, and 4A-cMDI), and thus to be neutrophils. However, in general it is very difficult and a matter of debate to discriminate the two populations [1,2]. Also any additional functional testing has not to bring more clarity [1,2]. Thus, we use(d) the term neutrophils to describe this cell population but are aware that it is not the whole truth. Therefore, we added now the information that neutrophils and G-MDSC could be members of the population in the figure legend.
- Results from 4T1 data in which the number of CD8 cells is about 10% seem very high compared to other reports, and the ratio to CD4 cells also seem inconsistent. Can these discrepancies be addressed?
Regarding this point we have to thank you very much for your attentive proof reading, since my colleagues preparing the graph detected a striking CD8+ copy/paste error happened within the graph data processing. We have corrected it in the revised manuscript, so that the number of tumor infiltrating CD8+ goes into normal range, and thus also the CD4/CD8 ratio.
- Your proposal “It is important to compare the immune content of the original tumor to several later passages (or just several passages) to show how whether the immune contexture is maintained over multiple passages. Knowing whether this is true would make this a much more characterized model. Also, between tumored mice in the same passage (i.e. see models with divergent growth pattern within a cohort of mice).”: strikes at the heart, but would -in my opinion- go beyond the scope of these two manuscripts, which are principally describing establishment and general properties of the new MDI models. Thus, I think more detailed analyses of a single MDI, the comparison of various sMDI, cMDI or between them by flow cytometry, RNAseq analyses and e.g. possibly by immuno-histochemistry should be the matter of future investigations.
- The RNAseq section should be removed: We do not agree with you in this point. It is true that not much (useful) data could be presented from the very small window of analyzed expression of various genes in only three gene families by RNAseg. But already the few data demonstrate i) the very different gene expression patterns between two related sarcomas (JA-2011 and JA-2042) but differing in route (s.c. versus i.m.) and carcinogen (MNU versus MCA) used for induction. ii) Data also confirm flow cytometry data regarding various tumor infiltrating
leukocytes by RNAseq, e.g. high CD44 gene expression in M2-macrophage infiltrated JA-0009, or enhanced Cd4 gene expression in JA-2041 with actual enhanced CD4 cell infiltration, which justifies to my opinion the presentation of these results.
Thus, we revised this section to make clear our arguments, and referred more exact to these results (only presented in supplementary data of manuscript #1) also in manuscript #2. We introduced also Figures better summarizing these results. Additionally, we will discuss with the editors to make independently available the supplementary data for both manuscripts each.
- It is necessary to have extensive editing for proper English usage – misspelled - carefully proofread the paper: To improve entire readability we made use of the MDPI “Specialist edit” service. It is clearly better now, but if it was helpful at all – I don´t know – it was hard work again to eliminate some newly introduced mistakes, e.g. as “graft versus the host reaction”.

Thank you again for your work

Kind regards,

Peter

(on behalf of all coauthors)

1. Bronte, V.; Brandau, S.; Chen, S.-H.; Colombo, M.P.; Frey, A.B.; Greten, T.F.; Mandruzzato, S.; Murray, P.J.; Ochoa, A.; Ostrand-Rosenberg, S., et al. Recommendations for myeloid-derived suppressor cell nomenclature and characterization standards. Nature Communications 2016, 7, 12150.
2. Zilio, S.; Serafini, P. Neutrophils and granulocytic mdsc: The janus god of cancer immunotherapy. Vaccines (Basel) 2016,4

Reviewer 3 Report

Comment synopsis:  805 words

Manuscript ID: cancers-409369, entitled “Mouse-Derived Isograft (MDI) in vivo Tumor Models II. Carcinogen-Induced cMDI Models: Characterization and Cancer Therapeutic

Approaches” evaluated the capacity of carcinogens (3-Methylcholanthrene (MCA) or N-Methyl-N-nitrosourea (MNU)) to induce Experimental Mouse Cancer models, with tools of molecular, cellular, and histological, pharmacological, and in vivo assays.

They tried to elucidate these newly created models unavailable with standard syngeneic tumor models through analysis of time scale for tumor growth periods, take rates, appearance, earliest time onset, tumor morphology, FACS biomarkers, and the reaction to immunotherapy, with 34 references and 2 tables plus 5 Figures.

The Manuscript appear to be systematic, of intriguing and of interest. It’s novel in its angle of design; however, the current draft suffers from lack of clarity, compromising its quality for publication. Specific 22 Comments and Suggestions for Authors (below) should be incorporated for its clarity, coherence, and logic flow.

Specific Comments, Questions, and Suggestions for Authors:

Current title is awkward and confused in logic.

Abstract needs to be restructured to remove all references.

Page 1, Lines 22-23: “Growth 22 curves of four sarcomas showed striking heterogeneity.” – How did you measure the quantity of heterogeneity?

Line 24: “variable invasion of immune cells” – any quantification?

Lines 29-30: “the tumors contain conserved tumor characteristics and intratumoral immune cell populations.” - – any quantification of such comparison? Compared side-by-side-in-experiment with other non-carcinogen-induced models (spontaneous tumors (sMDI))?

Line 52: “These new spontaneous tumor models enable studying causes” – some details need to establish the comparison.

Line 64: “waiting for spontaneous tumors” – for how long? How much tumor?

Line 66: “tumors are mostly sarcomas” – what types of another tumor? % of others? Consistency?

Lines 82-83: “3 animals died within 1-2 days after injection for unknown reason.

” – any speculation? ¾ died – high percentage alarming. Why no i.p. for female mice?

 Line 428: “To amplify the number of tumor pieces derived from one primary tumor, tumor pieces” – any criteria to choose which tumor piece? What size? As these two relate to future tumor growth and relapse, you can refer to Cancer genomic research at the crossroads: realizing the changing genetic landscape as intratumoral spatial and temporal heterogeneity becomes a confounding factor. Li SC, Tachiki LM, Kabeer MH, Dethlefs BA, Anthony MJ, Loudon WG. Cancer Cell Int. 2014 Nov 12;14(1):115. doi: 10.1186/s12935-014-0115-7. eCollection 2014. PMID: 25411563

 Lines 91-92: “all other established cMDI tumors were identified as sarcomas” – any explanation?

 Figure 2: 5 males and 3 females – any sex factors play a role in Histopathological characterization of established cMDI tumors? Arrow-head should be used to locate typical patterns.

  Line 112: “the total observation time of about 10 months three other tumors were detected” – what was the reason to keep so long?

 Line 119: “displayed identical pathohistological properties (Fig. 2, Tab. 2).” – Can you elaborate specific criteria being used to justify “identical” ? How many view fields being applied with what scale of measurement? Identical is a more quantifier term preferred. As shown on pages 5-6, you provided morphological descriptions. E.g., line 136: “numerous giant mononuclear or multinucleate giant cells” - numerous is not a quantifier.

 Line 120: “tumors or conspicuous tissues were detected in a female mouse” – Can you define what’s tumor and what’s conspicuous tissue with images to illustrate? Why only present in 2 females?

 Line 154: unexpected death. Any autopsy analysis to figure the cause of unexpected death?

 â€śtime scale for tumor growth periods, take rates, appearance, i.e. earliest time onset of tumor” – Figure3: “Growth curves of 10 to 12 mice” – which panel had 10 or 12 mice? It seems none of these four panels had 10 or 12? Replot the data with error bar (SD value), which can tell the trend preferred.

  Figure 4A – poor resolution – can’t see anything. Specific conclusive numbers should be indicated.

 Figure 4B: all with CT26-WT show low value – were the cells low capacity of tumor induction? How old were the cell lines? Any double labeling for these samples? How much percentage of were double-positive, triple positive?

 Figure 5: why JA-2019 with only 21 days? With huge difference of vehicle from others?

 Lines 302-304: “Whereas sMDI comprise adenocarcinomas, lymphomas, or histiocytic sarcoma/histiocyte associated lymphomas [1] the predominant tumor entity of cMDI were sarcomas.” sMDI model is more heterogeneous than cMDI – any explanation? How consistent could be seen with either model given such different scale of heterogeneity? How much relevant to human conditions?

 Lines 353-356: “First preliminary RNA-sequencing data regarding expression of tyrosine kinase receptors, IFN-353 Îł-signature or immune cell population markers showed various patterns comparing the “non-354 responder” JA-2011 and “weak-responder” JA-2042 models which are shown in the accompanying 355 paper in Tab. S1a, b, c [1]” – I didn’t see any supplements.

Author Response

cMDI - Author’s Reply to the Review Report - Reviewer3

Dear Madam, dear Sir,

Thank you very much for critical reviewing and your helpful suggestions for revision of our manuscript cancers-409369 “Mouse-Derived Isograft (MDI) in vivo Tumor Models II. Carcinogen-Induced cMDI Models: Characterization and Cancer Therapeutic Approaches”.
First of all, your remark “Current title is awkward and confused in logic” was elusive to us. In our understanding the title well describes the idea of propagating intact tumor pieces –in analogy to PDX- in syngeneic mice (isografts) as valuable term(s) of having model(s) with a more original tumor microenvironment, including immune components, for analyzing e.g. cancer immunotherapies as well as other therapies. Recommended restructuring of the abstract probably might result from a misunderstanding, since the faulted numbers in round brackets in the abstract are referring to respective numbers of various established tumors but not to any literature reference. It is now a little bit changed and unambiguously.
Coming back to your remarks regarding “quantity of heterogeneity” (Line 22-23), “invasion of immune cells” (Line 24), as well as “conserved tumor characteristics and intratumoral immune cells” (Line 29-39) we would like to give two answers. First, these points cannot be extended in the abstract by the word limit of a maximum of 200 words. Second, heterogeneity was not directly compared between various models but is well documented e.g. by heterogeneous growth curves (Fig. 2) as well as the dot plots of individual mice in the vehicle control group(s) (Fig. 5). For comparison, a more homogeneous growth can be seen for example in the LL/2, the 4T1, or the RENCA syngeneic standard mouse tumor models (Fig. S2-sMDI – supplementary data of accompanying paper). We will try to give you access to these data by the editor. Although it seems to be possible in principle also to quantify this heterogeneity (e.g. by the standard deviations of individual tumor volumes at a given time point), we don´t feel that this would actually enhance comparability of the models. Invasion of immune cells as well as conserved tumor characteristics and intratumoral immune cells was mainly assessed by pathohistological observations (TL), and partially confirmed by flow cytometry or RNAseq analysis. But we did not intend an actual quantitative comparison.
We summarized in Fig. 2 the relevant histopathological data as origin of primary isolate, H&E staining of this tissue and of another, s.c. re-transplanted follow-up-sample of the same tumor, together with tumor name, mouse strain, sex, and the tumor diagnoses, in order to submit as much information as possible in one figure. I think, in general the figure is very informative, but not unusable to give more detailed information. However, all further information and histological details could not be included into Fig. 2 (e.g. regarding to your comment to Fig. 2). Thus, we have now strikingly enhanced histological documentation in supplementary files (see revised manuscripts).
Regarding your questions to the spontaneous appearing sMDI models. We did not go into detail in this manuscript since two ones are planed as tandem paper, and writing the manuscripts we tried to reduce redundant information. But some of it, however, are already contained, e.g. in Line 42, nine tumors, and line 52, adenocarcinomas, lymphomas, or histiocytic sarcoma/histiocyte-associated lymphoma, in the manuscript. Please contact the editors to get access also to the other manuscript, i.e. to more information to sMDI (we will ask, too).
Concerning your comments:
Line 82-83: It was actually surprising and alarming. Yes. We started i.p. application only in male animals surprisingly and alarmingly 3/4 animals died within 1-2 days. The fourth one, however, was well and did not show any impairments. Since we really did and still do not have any idea why the animals died, we did not treat the female mice via the i.p. route, i.e. this group was removed from the study. One speculation –but not verified- could be that a different carcinogen compatibility caused the death (original dosage data were described for BALB/c or C57BL/6) in the CBA/J mice. Thus, we did not treat any female mice via the i.p. route.
Line 428: For details of implantation and amplification please see Line 599- 611
Line 91-92: Some carcinogens, e.g. MCA (and MNU), seem preferentially to induce sarcomas at least in some mouse strains.
Figure 2/Table 2/Line 119/Line 136: As written above, we extended histological documentation in the supplementary data of the revised manuscript(s) showing more detailed histological findings (e.g. marked by arrow-heads) in various tissues. But we do not believe that there are any sex factors or differences do influence neither carcinogen-induction nor histopathological outcome of cMDI tumors. Only 1/5 male tumors was a carcinoma (versus seven sarcomas) and the surplus of male to female tumor of 5:3 was only seen in established tumors, whereas preliminary isolated tumors or conspicuous tissues (Line 123 – 130) showed with 4 : 5 in male : female relation a rather opposite image.
Line 112: The goal of the study was to establish greatest possible number of carcinogen-induced tumor models obtained also by a prolonged observation period.
Line 120: Line 123-130 describe preliminary isolated tumors or conspicuous tissues in nine further mice (4 male; 5 female), “not yet finally established” (Line 128-130). The term was removed and these isolates were now only short accounted as potential MDI samples in the text or legend of tables to complete the number of all observed malignancies. Conspicuous tissues were any tissues (e.g. lymph nodes, thymus, and salivary glands, uterus stomach, or sternum) which not really appeared as tumors but macroscopically differed in size, form, or appearance from their “normal” counterparts, which were assumed as potential malignant candidate tissues.
Line 154: All dead animals were autopsied but in so called “unexpected death” macroscopically no obvious death causes could be identified.
Fig: 3: 10 mice (JA-2041) or 12 mice (JA-2011, -2019, and -2042) mice developed growing tumors (see take rate Tab. 1). Each individual curve represents an individual mice (no error bars) and its tumor volume. In some cases (see in JA-2011 and JA-2041) individual tumors/mice did not grow longer than 10-15 days (lost by unexpected death or ethical reasons). Thus, it appears that none of the panels would show 10 or 12 animals/tumors. This problem is reflected also by the so called Real Running Time (RRT) which mirrors actually available (alive) animals during a distinct observation period (Tab. 2).
Fig. 4a: We enhanced caption of single subpopulations
Fig. 4b: CT26wt cells showed low levels (but similar to most other cell line tumors) only in case of CD4+, CD8+ cell, or neutrophil infiltration, whereas in case of M1-, M2-macrophages or M-MDSC cells a rather moderate infiltration was observed. All flow cytometry staining are multicolor ones with gating by using FMO controls (fig. 4a).
Infiltration and tumor induction are not connected, neither in CT26-wt nor other cell line models (see supp. data – accompanying paper).
Fig. 5: As already shown in Tab. 2 JA-2019 was most “sensitive” tumor model regarding RRT (only 75 % alive animals on day 21) with only 7/10 animals in the vehicle group in this experiment. On the other hand it was the only tumor which was sensitive to anti-PD1 or anti-CTLA4 mono-therapy. It will be the matter of further studies to define causes for these difference(s). See also revised discussion
Line 302-304: It was (also) the matter of the two paper to “discriminate” between spontaneous appearing (sMDI) and chemical induced tumors (cMDI). It was found –as expected- that spontaneous MDI tumor entities rather correspond to spontaneous tumors in patients, whereas cMDI are preferentially carcinogen-induced sarcomas.
Line 353-356: As written above supplementary data have been only part of the accompanying paper. They are now extended, and we will try to give you access to these data by the editor, too.
We hope revised manuscript will answer also in these points your´s expection for possible improvement of the manuscript. Thank you again for your support.
Thank you again for your work
Kind regards,
Peter
(on behalf of all coauthors)
Peter Jantscheff PhD (group leader)
In Vivo Pharmacology
Proqinase GmbH
Breisacher Str. 117
79106 Freiburg
Germany
peter.jantscheff@t-online.de
Tel: +49-7666-913-0396

Round  2

Reviewer 2 Report

The authors have addressed my major concerns sufficiently.